# Effect of Sintering Temperature and Polarization on the Dielectric and Electrical Properties of La_0.9_Sr_0.1_MnO_3_ Manganite in Alternating Current

**DOI:** 10.3390/ma15103683

**Published:** 2022-05-20

**Authors:** Wided Hizi, Malek Gassoumi, Hedi Rahmouni, Ahlem Guesmi, Naoufel Ben Hamadi, Essebti Dhahri

**Affiliations:** 1Unité de Recherche MatériauxAvancés et Nanotechnologies (URMAN), Université de Kairouan, ISSAT, Kasserine 1200, Tunisia; wided@live.com (W.H.); rahmounhedi@yahoo.fr (H.R.); 2Chemistry Department, College of Science, IMSIU (Imam Mohammad Ibn Saud Islamic University), Riyadh 11623, Saudi Arabia; amalkasme@imamu.edu.sa (A.G.); nabenhamadi@imamu.edu.sa (N.B.H.); 3Laboratoirede Physique Appliquée, Faculté des Sciences, Université de Sfax, Sfax 3000, Tunisia; desebti@yahoo.com

**Keywords:** La_0.9_Sr_0.1_MnO_3_, sintering temperature, electrical conductivity, hopping, tunneling, polarization and dielectric properties

## Abstract

The electrical characterization ofa La_0.9_Sr_0.1_MnO_3_ compound sintered at 800, 1000 and 1200 °C was investigated by means of the impedance-spectroscopy technique. As the results, the experimental conductivity spectra were explained in terms of the power law. The AC-conductivity study reveals the contributions of different conduction mechanisms. Indeed, the variation in the frequency exponents (‘s_1_’ and ‘s_2_’) as a function of the temperature confirms the thermal activation of the conduction process in the system. It proves, equally, that the transport properties are governed by the non-small-polaron-tunneling and the correlated-barrier-hopping mechanisms. Moreover, the values of the frequency exponents increase under the sintering-temperature (TS) effect. Such an evolution may be explained energetically. The jump relaxation model was used to explain the electrical conductivity in the dispersive region, as well as the frequency-exponent values by ionic conductivity. Under electrical polarization with applied DC biases of Vp = 0.1 and 2 V at room temperature, the results show the significant enhancement of the electrical conductivity. In addition, the dielectric study reveals the evident presence of dielectric relaxation. Under the sintering-temperature effect, the dielectric constant increases enormously. Indeed, the temperature dependence of the dielectric constant is well fitted by the modified Curie–Weiss law. Thus, the deduced values of the parameter (γ) confirm the relaxor character and prove the diffuse phase transition of our material. Of note is the high dielectric-permittivity magnitude, which indicates that the material is promising for microelectronic devices.

## 1. Introduction

The physical properties of manganites have been extensively studied for the last several years [1,2,3,4,5,6,7,8]. These rich properties allow us to exploit these types of materials in several applications [1,2]. Furthermore, the feature that makes this material more useful is the ability to control their structure and their electrical and magnetic properties by many ways. Indeed, the doping, whether in the A or B sites or both, and the sintering temperature (TS), with all of their circumstances, as well as the elaboration process, are the most influential parameters for the enhancement of the behavior of these materials. These treatments influence the Mn^3+^–O_2_^−^–Mn^4+^ network, which is responsible for the electrical conduction in manganites. As mentioned above, the sintering temperature can affect this network, in which the rise in the TS causes an increase in the grain size, which, in turn, affects the double-exchange (DE) interactions [2] between the Mn^3+^ and Mn^4+^ ions that reach the grain-boundary region. In the literature [1], this thermal excitation modifies the metal–semiconductor-transition temperature. In addition, it improves the homogeneity and the crystallinity of the samples, and it provides a better connectivity of the grains. In this context, lanthanum manganite systems are widely studied by using the effects of the previously noted parameters [2]. Assoudi et al. [9] report that the existence of electric polarizations can be identified by the permittivity spectrum. In addition, from this spectrum, the colossal static dielectric constant makes the doped lanthanum manganite act as a good candidate for different applications in electronic industrial fields. For the La_0.6_Sr_0.2_Na_0.2_MnO_3_ compound, the behavior of the dielectric constant is governed by the Maxwell–Wagner theory of interfacial polarization [10].

Out of all the doped lanthanum manganites, numerous research groups are interested in the assessment of the physical properties of lanthanum strontiummanganites (LSMO) [11,12,13,14,15,16,17,18,19,20,21]. In fact, LSMO systems can act as a cathode in solid oxide fuel cells because of their good electrical conductivity [18]. Moreover, they are used in medicine and in electronic industrial fields [19,20,21], owing to their colossal magnetoresistance [22] and magnetocaloric [23] effects. Different technological applications are based on the high dielectric permittivity of this material (in the order of 105 [2]). Since the dielectric properties are related to the electrical ones, it is necessary to comprehend the conduction process that is present in the investigated material. However, the partial substitution of the divalent ion Sr^2+^ in the parent compound LaMnO_3_ on the lanthanum site produces the creation of the mixed valence for the manganese that leads to mobile charge carriers. Usually, the movement of these charge carriers can be manifested by the hopping and tunneling processes. In this context, different theoretical models can be used to explain the conductivity spectrum [2,24,25,26,27,28]. 

The magnetic study on our material, La_0.9_Sr_0.1_MnO_3_, sintered at 800, 1000 and 1200 °C was investigated as reported in [23,29,30]. A paramagnetic–ferromagnetic phase transition at the Curie temperature (TC) was observed for all the sintered samples. Moreover, this Curie temperature was found to increase with the reduction in the particle size. Otherwise, the magnetocaloric properties reveal the possibility of extending the application of a random magnetic anisotropy model, which was initially developed for amorphous alloys, to nanocrystalline materials. Moreover, our studied compound can act as a candidate for magnetic refrigeration because of itsrelative cooling-power values, which are close to those of commercial magnetic refrigerant materials [30]. To complete the electrical side of the whole study, we started the exploration of the transport properties of the La_0.9_Sr_0.1_MnO_3_ system [31]. To the best of our knowledge, the dielectric properties of the LSMO system have not been well studied. In this paper, we present the effects of the sintering temperature and polarization on the dielectric and electrical properties of the La_0.9_Sr_0.1_MnO_3_ system by using the impedance-spectroscopy technique.

## 2. Experimental Details

The La_0.9_Sr_0.1_MnO_3_material was synthesized by using the citrate–gel method. In fact, stoichiometric amounts of the nitrate precurs orreagents La(NO_3_)_3_ 6H_2_O, Mn (NO_3_)2 4H_2_O and Sr(NO_3_)_2_ were dissolved in water and were mixed with ethylene glycol and citricacid, which formed a stable solution. The molar ratio of metal: citricacid was 1:1. The solution was then heated on a thermal plate under constant sintering at 80 °C to eliminate the excess water and to obtain a viscous gel. The gel wasdried at 130 °C, and it was then calcinated at 600 °C for 12 h, and was sintered at 800, 1000 and 1200 °C, as detailed in [29]. The structure and phase purity of the powders were verified by X-ray diffraction (XRD) by using Cu–K_α1_ radiation at room temperature [29]. The XRD patterns reveal that all samples are single phase, with no detectable secondary phases. Indeed, the samples thatannealed at 800 and 1000 °C were found to be crystallized in the rhombohedral structure, which is attributed to the *R3c* space group, whereas two distinct phases were observed for the sample sintered at 1200 °C, in which it was crystallized in the rhombohedral and orthorhombic structures that belong, respectively, to the *R3c* and Pbnm space groups. From the X-ray-peak width and by using Scherrer’s relation, the average particle sizes of the samples were assessed [29]. The TEM (transmission electron microscopy, JEOL 2010F-TEM, Santa Rosa, CA, USA) imaging confirms the obtained particle-size values from the XRD patterns (see Table 1 in [29]).

The powders obtained were ground and then pressed into pellets that were a few millimeter sthick (~2 mm), and with a diameter of around 1 cm under 10 tonnes/cm^2^. These pellets weresintered at 1000 °C for 24 h. In order to obtain the pure crystalline phases, the obtained pellets that were finely ground, and the resulting powders, underwent some further cycles of grinding, pelleting and sintering at 1100 °C. Then, on both sides of each pellet, thin silver films were evaporated by Joule heating through a circular mask. These silver films act as electrodes. The obtained electrodes were used to conduct the electrical characterization. Indeed, the capacitance (C), the dissipation factor (D) and the conductance (G) were measured by using an Agilent 4294 A impedance analyzer (Santa Clara, CA, USA) at various temperatures over a large frequency range (40 Hz–10 MHz). The electrical conductivity is deduced from the conductance and the geometric factors of the pellets, and the dielectric permittivity is determined from the capacitance (C), the dissipation factor (D) and the geometric factors. The samples were mounted onto aJanis VPF 800-cryostat (Strasbourg, France) to vary the temperature between 80 and 700 K by using liquid-nitrogen cooling. All measurements were investigated in darkness and under vacuum to avoid the impact of the illumination and the ambient atmosphere.

## 3. Results and Discussion

### 3.1. Frequency-Dependent AC-Conductivity Study

Figure 1a–c shows the electrical-AC-conductivity (σ_AC_) spectra at various temperatures for the La_0.9_Sr_0.1_MnO_3_ compound sintered at 800, 1000 and 1200 °C. As can be seen, the clear effect of the temperature is observed over the entire frequency range for each conductivity spectrum. This result proves that the conduction process in the investigated samples was thermally activated. These spectra are composed of two distinctive frequency regions, according to their behaviors. In fact, a plateau appears for each spectrum at low frequencies, in which the conductivity is practically frequency independent and thermally activated. This region is associated with the long-range translational motion of the charge carriers [32]. In the second frequency region, the electrical conductivity rises with the rise in the frequency, which shows the imprints of the semiconductor behavior and reflects the potential interventions of different conduction mechanisms. Such behavior is due to the creation of new conduction sites, and to the release of the trapped charge carriers under the frequency effect. This evolution confirms that the hopping and the tunneling processes are the most likelyexplanations to describe the electrical conductivity. In fact, the basic idea of AC conductivity is manifested by the fact that it is an increasing function, with the frequency for any type of hopping or tunneling process taken into account. However, this rise presents two linear variations, which are separated by a transition region, as is shown in Figure 1d. This is attributed to the presence of two frequency exponents for all the samples. The frequency-exponent values are determined from the slopes of these linear variations. Thus, different kinds of transport processes take place in the compound, and they depend on the frequency and the temperature. Such an observation leads to the fact that the AC conductivity follow to the Jonscher’s power law given by [32,33]:(1)$$\sigma_{AC}=\sigma_{0}+A\omega^{s_{1}}+B\omega^{s_{2}}$$
where Aω^s1^ and Bω^s2^ are, respectively, the high and the intermediate AC-conductivity responses (*σ*_0_); A and B are constants; ω is the angular frequency; and ‘s_1_’ and ‘s’ are the frequency exponents that indicate the degree of the interaction between the charge carriers in motion and the surrounding lattices [34]. This agreement between the experimental results and the theory is detected at temperature ranges of 80–500, 80–50 and 80–280 K, respectively, for the samples annealed at 800, 1000 and 1200 °C. Such behavior has been observed in manganite compounds [2].

For the sample annealed at 1200 °C, the conductivity can be governed by the universal “Jonscher Power Law” (σ_AC_ ∝ ω^s^) [35,36] in the temperature range of 300–600 K, where “s” is temperature dependent. Indeed, as the frequency increases, the σ_AC_ increases with a single linear variation beyond the observed plateau. The change in the slope, at which the σ_AC_ increases, is marked at a specific frequency that is known as the “hopping frequency”. The conductivity spectra obey the following expression [35,36]:(2)$$\sigma_{AC}=\sigma_{0}+A\omega^{s_{1}}$$

Then, for the 600 and 700 K temperatures, the samples that were sintered at 800 and 1000 °C exhibit conductivity decreases with the increase in the frequency beyond the detected plateau. Such a result informs us about a metallic behavior that is governed by the classical Drude model [37], whereas, for T_S_ = 1200 °C, this metallic behavior appears only for 700 K. Thus, the semiconductor behavior extends from 80 to 600 K. Additionally, this sample is also specialized by the appearance of a peak at high frequencies in its AC-conductivity spectrum. This peak is detected at a specific frequency that is known as the “relaxation frequency”. Then, as the frequency increases, the conductivity decreases, and all the temperature curves merge, which are temperature independent.

The temperature dependences of the frequency exponents (‘s_1_’ and ‘s_2_’) for all the sintered samples are depicted in Figure 2a–f. The observed variation in the ‘s’ values in this figure with the temperature confirms the thermal activation of the electrical-transport mechanism. It also proves that the fact hopping is the potential mechanism that governs the transport properties [37]. Indeed, the frequency exponent ‘s_1_’ was deduced from the dispersive part of the high-frequency region (Figure 1d). In Figure 2a,b, the ‘s_1_’ increases with the temperature increase in the first region of temperature. This proves that non-overlapping small polaron tunneling (NSPT) [38,39,40,41] is the most suitable model to characterize the conduction for T_S_ = 800 and 1000 °C. This increase in ‘s_1_’ with the rise in the temperature is in good agreement with the theory that is described by the following expression [38,39,40,41]:s = 1 + (4k_B_T/W_H_)(3)
where W_H_ is the polaronhopping energy and k_B_ is the Boltzmann constant. Then, ‘s_1_’ continuously decreases with the risein the temperature. Such a variation confirms the contribution of the correlated-barrier-hopping (CBH) mechanism in conduction [38,39,40,41]. However, this model (CBH) is marked, equally, from the variation in the exponent ‘s_2_’ with the temperature, which decreases in the entire temperature range (Figure 2d–f). This is obtained from the intermediate frequency ranges for all the samples (Figure 1d). The decrease in ‘s_2_’with the temperature increase for the CBH model is explained by the following relation [38,39,40,41]:s = 1 − (6k_B_T/W_M_)(4)
where W_M_ is the binding energy, which is the energy that is required to remove an electron from one site to the conduction band. Accordingly, the variation in the frequency exponent (‘s’) with the temperature comes from the fact that the effective frequency of the phonon that is involved in the polaron formation depends on the temperature [42]. An agreement with the literature of the found ‘s_1_’ behavior is reported by Moualhi et al. [2] for an LCAM compound. The authors observed the NSPT process from 80 to 170 K. Beyond 170 K, the conduction was governed by the CBH model [2].

For the sample that was sintered at T_S_ = 1200 °C, the exponent ‘s_1_’ values, which are represented in Figure 2c, exceed the unity. This indicates that the classic previous models cannot explain the charge-carrier transport in this temperature range. For this reason, the jump relaxation model (JRM) was used to explain this evolution in perovskites and amorphous materials [43,44,45]. This model tries to describe andvisualize the dynamics of the jumping motion as a function of space and time, where two opposing relaxation processes occur, which are successful jumps and unsuccessful jumps [46,47,48,49,50,51]. The physical significance of the JRM originated from the jumping motion of the ions in the atomic scale (ionic conductivity) [46,47,48,49,50,51]. According to the JR model, the displacement of the mobilized ions is explained in terms of the frequency effect. Furthermore, this model takes into consideration the interactions between a mobile ion and the neighboring defects. This model was proposed by Funke [46]. Indeed, Funke [46] explains the observed plateau, at low frequencies, by the successful jump to a vacant adjacent site that is thanks to the long period of available time. As the frequency increases, the ion is temporarily out of equilibrium with respect to the distribution of the neighbors. In this case, two concurrent relaxation processes can be visualized (successful and unsuccessful jumps) [43,44,45,46,47,48,49,50,51]. Indeed, for the successful jumps, when an ion jumps to a new site, its neighboring ions relax and redistribute to establish a new equilibrium condition around the new site. This generates the increase in the ACconductivity in the first dispersive region (the intermediate frequency range). Such an increase is associated with the short-range translational motion that results in the term Bω^s2^, where 0 < s_2_ ≤ 1. At high frequencies, the ion is forced to hop back to its initial position (since it is unable to stabilize energetically, and the available time is very short), and it performs a sequence of correlated forward–backward jumps [44,46]. This is the unsuccessful jump. The term Aω^s1^ characterizes, then, the dispersive region of high frequencies, which corresponds to the localized jumping or the reorientation movement of ions, where 0 < s_1_ ≤ 2. There is a greater probability that more jumps are unsuccessful in high frequencies (the second dispersive region) [44]. The dispersive conductivity is then generated by the increase in the ratio of successful to unsuccessful jumps.

According to Funke [46], if *s* ≤ 1, the hopping includes a translation motion that is accompanied by sudden hopping, while, in the case of *s* > 1, the hopping motion includes localized hopping between neighboring sites. In this context, it is observed that ‘s_1_’ and ‘s_2_’ increase with the rise in the T_S_ (Figure 3a,b). As can be seen from Figure 2a–f, the impact of this heat treatment is shown in the ‘s_1_’ and ‘s_2_’ values. It is also observed that the deduced energies decrease with the increase in the sintering temperature. Indeed, W_H_ values were obtained that were equal to 0.118 and 0.088 eV for T_S_ = 800 and 1000 °C, respectively. The energy value for T_S_ = 800 °C (0.118 eV) is in good agreement with that of the lanthanum manganite system (0.115 eV) [2]. The W_M_is decreased from 0.264 to 0.197 eV with the increasing T_S_. The same evolution was also detected for the W_M1_ and W_M2_ that were extracted from the [1-s_2_] temperature dependence. Such results can be related to the fact that the hopping motion becomes localized between neighboring sites as ‘s_1_’ increases, until it exceeds «1» for T_r_ = 1200 °C (JRM) [46]. Thus, the charges carriers need lower energy to jump between sites, since the hops occur over shorter distances. In addition, the jumps become realized towards higher levels [46,47,48,49,50,51] for T_S_ = 1200 °C (JRM), which is in contrast to the horizontal translational motion in the classic models for T_S_ = 800 and 1000 °C. Such a result shows the sensitivity of the conduction mechanism to the sintering temperature. Furthermore, the values of ‘s_2_’ are very low for T_S_ = 800 and 1000 °C, which indicates that a hard hopping process requires high energy (Figure 2d,e), whereas, for T_S_ = 1200 °C, the ‘s_2_’ values increase and vary between 0 and 1, and the energy values decrease, as is shown in Figure 2f. Thus, the sintering temperature promotes the hopping process.

The W_H_, W_M_, W_M1_ and W_M2_ were calculated by using Equation (3) (for W_H_), Equation (4) (for W_M_, W_M1_ and W_M2_) and the obtained slopes from the linear fit of [1-s_1_] (for W_H_ and W_M_) and [1-s_2_] (for W_M1_ and W_M2_) against the temperature.

### 3.2. Polarization Effect

At room temperature, the same measurement was conducted under electrical polarization with applied DC biases of V_p_ = 1 and 2 V. Figure 4a–c shows the conductivity spectra of the La_0.9_Sr_0.1_MnO_3_ material that was sintered at 800, 1000 and 1200 °C. All the spectra show a considerable increase in the conductivity as the V_p_ increases. Such a result elucidates evidence of the space-charge contribution in the conduction. The applied electrical field helps with the scattering of the free charge carriers that are emitted by the trap centers across the space charge, which, in turn, enhances the conductivity [38,52]. This observation describes the effect of colossal electro-resistance, which makes our material a promising candidate for technologies of the future. This effect has been observed in perovskite materials [53,54]. According to Prakash et al. [55], the presence of the space-charge concept and the increase in the conductivity under polarization prove the contribution of the grain-boundary region to the electrical conduction. The authors report the dependence of the grain boundary on the variable DC bias, while the grain is independent. The observed grain-boundary behavior is explained by the grain-boundary double Schottky barrier [55].

### 3.3. Dielectric Properties

#### 3.3.1. Frequency Dependence

The dielectric measurements were conducted by the impedance-spectroscopy technique. The dielectric data are expressed in complex form: ε* = ε′ − jε″, where ε′ and ε″ describe, respectively, the stored and the dissipated energies. The ε′ and ε″ were calculated by using the following relations: ε′ = (C.t)/(ε_0_.A) and ε″ = ε′.D, where ε_0_ is the permittivity of the free space, A is the electrode area and t is the thickness of the samples. The frequency dependences of the real (ε′) and imaginary (ε″) parts of the dielectric permittivity at different temperatures for the La_0.9_Sr_0.1_MnO_3_ compound that was sintered at 800, 1000 and 1200 °C are depicted in Figure 5a–c and Figure 6a–c. Their behavior with the temperature and frequency have been spotted in perovskite materials [4,5,56,57,58,59]. At low frequencies, the ε′ shows very high values for all the samples, with a good stability for the sample sintered at 1200 °C (Figure 5a–c). Then, it decreases with the frequency increase, and it shows a dispersive behavior at high frequencies, where the ε′ values coincide with all of the explored temperatures. Such behavior can be related to the presence of different types of polarizations in the investigated samples, such as electronic, dipolar, ionic and interfacial polarizations [4,5,26]. At low frequencies, all four types contribute to the total polarization. The contributions of the interfacial and the dipolar polarizations dominate in the low-frequency range [60,61]. These two kinds of polarizations are strongly dependent on the temperature, which explains the temperature dependence of the ε′ [60,61]. However, the polarization process tends to follow the applied electric-field direction in this frequency range. Therefore, the maximum value of the ε′ is reached. Then, the rapid decrease in the ε′ can be explained by the increase in the free-charge-carrier density, which causes a reduction in these polarization kinds with the increase in the frequency [4]. Moreover, the difficulties that are encountered by the dipolar material in following the electrical-field direction can be an explanation for this decrease. This behavior has been observed for the La_0.7_Sr_0.3_Mn_0.8_Fe_0.2_O_3_ system [62]. At higher frequencies, the electronic and ionic polarizations contribute, in which the ε′ merges and becomes temperature independent. This independence arises from the negligible contributions of the interfacial and dipolar polarizations.

Hence, the high values of the ε′ may partially refer to the existence of space-charge zones that are produced by a localized accumulation of the charges on the electrode-sample contact and on the interface between the grains and the grain boundaries [3,4]. This is attributed to a Maxwell–Wagner (MW) type of interfacial polarization [63,64]. According to the MW [63,64] theory of interfacial polarization, the high values of the dielectric constant in the system can be related to the presence of grains and grain boundaries (a heterogeneous structure). Thus, the dielectric structure of the compound is composed of two phases: the first one is a conductive layer that consists of grains that are separated by the second layer, which is composed of poor conductive grain boundaries, as is shown in Figure 5d. Thus, the drop in the ε′ values was established when the charge-carrier hopping could not follow the applied field beyond a certain high frequency, which, relatively, prevents the fact that the electrons reach the grain boundary. Thus, the interfacial polarization is decreased, which allows us to conclude the decisive role of the grains and the grain boundaries in the transport properties in both low- and high-frequency ranges.

The frequency dependence of the imaginary part of the dielectric permittivity (ε″) at different temperatures is characterized by the appearance of a peak at a specific frequency (“f_res_”) for all the samples (Figure 6a–c). At this specific frequency, the dielectric material has the least stored energy in the maximum, which is shown in the ε″ spectra. Furthermore, the appearance of this peak can be explained by the proximity of the frequency hopping of the charge carriers to that of the applied field. Such behavior is defined by the resonance phenomenon [4]. For all the ε″ spectra, it is well observed that this peak shifts toward the higher frequencies as a function of temperature. This shift confirms the presence of the dielectric relaxation in the material [4,5]. The presence of this phenomenon is also proven by the shift in the observed drop in the real part of the permittivity to higher frequencies when the temperature rises. Moreover, a second relaxation peak is observed (Figure 6a–c). Such a result confirms the presence of a second relaxation process.

Accordingly, the dielectric behavior of the La_0.9_Sr_0.1_MnO_3_ compound demonstrates a Debye-like relaxation, which can be explained by Maxwell–Wagner polarization [63,64]. This model explains the synchronization of the dielectric-constant decrease and the AC-conductivity increase as the frequency increases (Figure 7). At low frequencies, the conductivity is frequency independent. Beyond the hopping frequency, the conductivity starts to increase, and it depends strongly on the frequency, following the Double Jonscher Power Law, whereas, in the frequency range at which the σ_AC_ increases, the ε′ shows two sharp drops, each one of which starts beyond a frequency-independence variation (Figure 7). Thus, the decrease in the permittivity is related to the conductivity increase that is induced by the rise in the number of charge carriers hopping. Such behavior underscores the strong correlation between the conduction and the dielectric permittivity [4,65,66,67]. This can be theoretically confirmed by the relations between the real and imaginary parts of the permittivity, the dielectric loss tangent (tgδ) with the electrical conductivity and the angular frequency (ω), as is shown in the following equations [4,10,66]:(5)$$\mathrm{tg}\delta =\frac{\sigma_{\mathrm{AC}}}{\omega \varepsilon_{0}\varepsilon^{\prime }}$$
(6)$$\varepsilon^{\prime \prime }=\frac{\sigma_{\mathrm{AC}}}{\varepsilon_{0}\omega }$$

These relations prove that the ε′ and the tgδ are inversely proportional to the ω, since the ACconductivity is directly proportional to the angular frequency (ω). As a definition, the dielectric loss tangent (tgδ) quantifies the inherent dissipation of the electromagnetic energy for dielectric materials, andit occurs when the polarization shifts behind the applied electric field. It is also related to the relaxation process and is defined as tgδ = ε″/ε′. The frequency dependences of the tgδ and the ε″ are represented in Figure 7 for the sample that was sintered at 1200 °C, andthey exhibit two dielectric-relaxation peaks. Thanks to the low values of the loss tangent and the high values of the dielectric constant, our material may be a promising candidate for tunable capacitors [5]. Similar results are reported by Rahmouni et al. [4,5].

#### 3.3.2. Temperature Dependence

For all the sintered samples, the evolution of the ε′ as a function of temperature for some selected frequencies is plotted in Figure 8a–c. The studied samples exhibit a dielectric permittivity that ranges between 10^3^ and 5×10^6^, which are considered to be high values. Thesehigh values make our compound, La_0.9_Sr_0.1_MnO_3_, a beneficial material for technological applications, such as microelectronic devices [62,68]. A focus on their evolution with the frequency confirms the decrease in the ε′ with the frequency increase for all of the samples (Figure 8a–c). For T_S_ = 800 and 1000 °C, the material presents two transitions at T_d1_ and T_d2_. Usually, T_d_ is designated as the dielectric-transition temperature [69]. Indeed, the ε′ decreases until reaching T_d1_. Then, it starts to increase with the temperature rise. Such behavior is mainly due to the increasesin the dipolar and interfacial polarizations when the temperature increases, which bring a thermal energy to the material [60,70,71]. Afterward, it realizes a maximum at a specific temperature (T_d2_). At this temperature, the hopping frequency of the electrons between Mn^4+^ and Mn^3+^ is equal to the frequency of the applied field. Beyond this temperature (T_d2_), the polarization no longer contributes, and the relative permittivity begins to decrease. Moreover, it is well observed that these transitions shift to lower temperatures for T_d1_ and to higher temperatures for T_d2_ with the increase in the frequency. These mentioned shifts emphasize the relaxor behavior. It is also marked that the first dielectric transition that was detected at T_d1_ disappears as the frequency rises. For T_S_ = 1200 °C, a decrease in the ε′ is observed, which is followed by the appearance of an anomaly peak that is marked at 400 K (Figure 8c). An analogous anomaly has been detected for manganite-type perovskite materials [70,71]. This peak may be ascribed to a local polarization, and it indicates the diffuse character of a phase transition in the material.

Figure 8d shows the temperature dependence of the dielectric constant at different sintering temperatures. It is observed that the ε′ increases with the sintering-temperature increase. This can be related to the polarization that is influenced by this thermal treatment. In perovskites, the elaboration method and the sintering temperature have a significant influence on the electrical properties. In our case, the samples are prepared by the citrate–gel method. The sintering temperature at 1200 °C is then considered high. This shows thatoxygen deficiencies that are near to the grain boundaries can be created in this sample [72,73,74]. Various studies have shown that oxygen defects contribute to the dielectric response. This is manifested in the reorientation of elementary electric dipoles [72,73]. The ionization of oxygen defects creates conduction electrons that could be trapped by Mn^4+^ cations, which leads to the formation of Mn^3+^ cations [74]. Then, defect dipoles are produced [74]. This causes the increase in the dielectric-constant values. These electrons can become conductive electrons by thermal activation. The dielectric behavior will be dependent on oxygen-defect ionization, and it leads to strong dielectric properties that are dependent on the frequency and the temperature [75]. In summary, the formation of oxygen defects is electronically compensated by changes in the oxidation state of the manganese (Mn) cations. This behavior has been observed in the literature for perovskite-type materials [74,75,76,77,78]. In addition, the increase in the unit cell volume, which is mentioned in the previous structural study [29] and which is due to the particle-size increase, can reduce the hops throughout the Mn^3+^–O^2−^–Mn^4+^ network. This minimizes the formation of Mn^4+^ ions. Therefore, the possibility of Mn^3+^ formation is greater, which is responsible for the increase in the polarization. This can also contribute to the increase in the dielectric constant. In fact, several studies have correlated this behavior to the particle-size effect [78,79,80]. Indeed, the Mn^3+^ concentration is more important for the bigger particles. Thus, the dielectric-permittivity magnitude is more important as the particle size increases. Equally, such a result can be due to the appearances of the grain boundary and the electrode effects, which constitute structural discontinuities. Therefore, the interfacial polarization between the grains and their boundaries, and between the electrodes and the sample, is a primary factor that influences the dielectric constant of the material.

According to the Curie–Weiss law, the dielectric constant (ε′) above the Curie temperature can be described by the following relation [61]:1/ε′ = (T − T_CW_)/C_CW_ (T > T_C_)(7)
where C_CW_ is the Curie–Weiss constant, and T_CW_ is the Curie–Weiss temperature that describes the temperature from which the ε′ starts to deviate from this law. The variation in the 1/ε′ as a function of temperature and performed at 10 kHz is plotted in Figure 9a–c. As can be seen, the ε′ is fitted by Equation (7) for all the sintered samples. Knowing that the T_m_ is the temperature that corresponds to the maximum of the ε′, it is observed that the T_CW_ value is greater than the T_m_ for each sample. Such a result suggests the diffuse phase transition and confirms the relaxor character of the La_0.9_Sr_0.1_MnO_3_ system.

The modified Curie–Weiss law was proposed by Uchino and Nomura [81], and it is described by the following relation:(8)$$\left(\frac{1}{\varepsilon^{\prime }}\right)-\left(\frac{1}{{\varepsilon^{\prime }}_{m}}\right)={\left(T-T_{m}\right)}^{\gamma }/C$$
where ε′_m_ is the maximum of the ε′, and C is a constant. The parameter γ gives a good idea about the character of the phase transition. For this reason, the insets of Figure 9a–c show the plots of ln[(1/ε′) − (1/ε′_m_)] against ln (T − T_m_) at 10 kHz. The linear fit that is depicted in these insets is used to estimate the γ values. In fact, the classical Curie–Weiss law is confirmed for γ = 1 and for γ = 2, and it describes a complete diffusephase transition [68]. For each curve, the γ values are found to be γ > 1 and γ ≈ 2 (γ = 1.69, 1.67 and 1.92 for T_S_ = 800, 1000 and 1200 °C, respectively). Such values prove that our material has a diffuse-type phase transition. Moreover, the obtained γ values support the relaxor nature of the LSMO system. Such an observation has been shown for an LCMO–Ag compound with γ = 2.09 [5]. The same phenomenon is observed from the temperature dependence of the imaginary part of the permittivity (ε″) at various frequencies for T_S_ = 1200 °C (Figure 10). Indeed, the examination of this variation confirms the relaxor behavior, in which the observed peak shifts to higher temperatures as the frequency rises.

## 4. Conclusions

The electrical and the dielectric properties of a La_0.9_Sr_0.1_MnO_3_ compound sintered at 800, 1000 and 1200 °C were investigated. The increase in the sintering temperature affects the electrical behavior of the material. The Double Jonscher power aw explains the coexistence of two linear regions in the conductivity spectra by the existence of two frequency exponents. As the temperature rises, the frequency-exponent values change. This change indicates that the hopping and the tunneling are the dominating transport processes in the material. Moreover, the agreement between the temperature dependence of ‘s_1_’ and ‘s_2_’ and the theory proves the contributions of the NSPT and CBH models in the conduction process. The rise in the TS produces an increase in the values of ‘s_1_’ and ‘s_2_’, whereas they maintain their behavior shape under the effect of this thermal excitation. The jump relaxation model explains the observed values of ‘s_1_’ for T_S_ = 1200 °C. Such results show the dependence of the transport properties on the sintering process. Furthermore, the results show that the σ_AC_ increases significantly when the Vp increases from 0 to 1, and then to 2 V. The dielectric study reveals the suitability of our samples to microelectronic devices. Their behavior was correlated to the polarization process. All the sintered samples behave in a similar manner to relaxors. A significant increase in the dielectric constant is observed when the sintering temperature rises.

## Figures and Tables

**Figure 1 materials-15-03683-f001:**
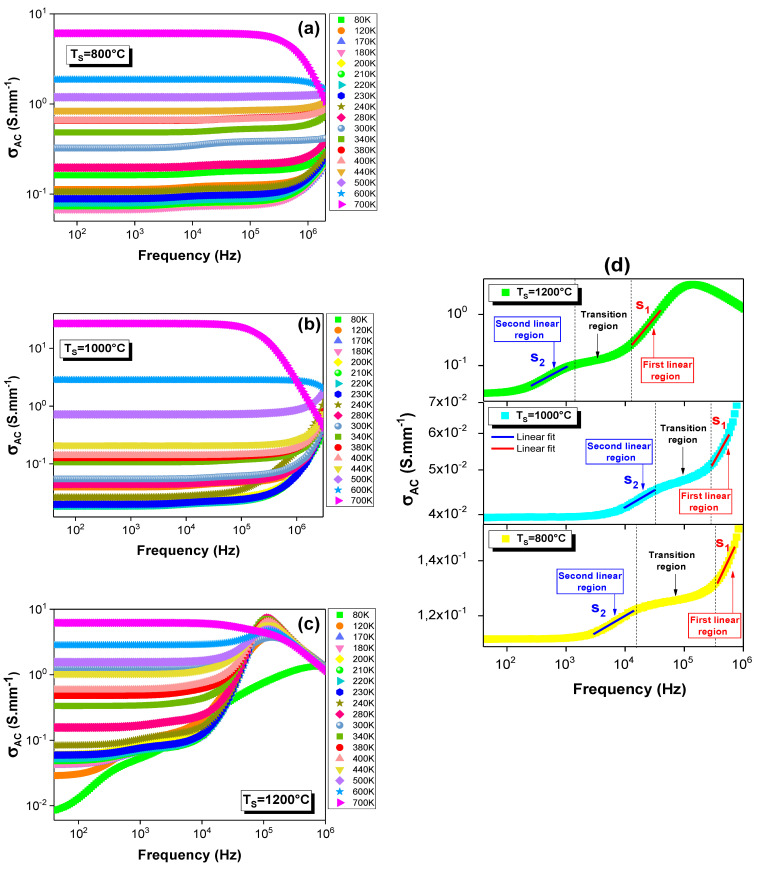
Frequency dependences of the electrical conductivity (σ_AC_) at various temperatures for the La_0.9_Sr_0.1_MnO_3_ system sintered at (**a**) 800, (**b**) 1000 and (**c**) 1200 °C. (**d**) Evolution of σ_AC_ as a function of frequency fitted and performed at 120 K for the La_0.9_Sr_0.1_MnO_3_ system sintered at different temperatures (TS).

**Figure 2 materials-15-03683-f002:**
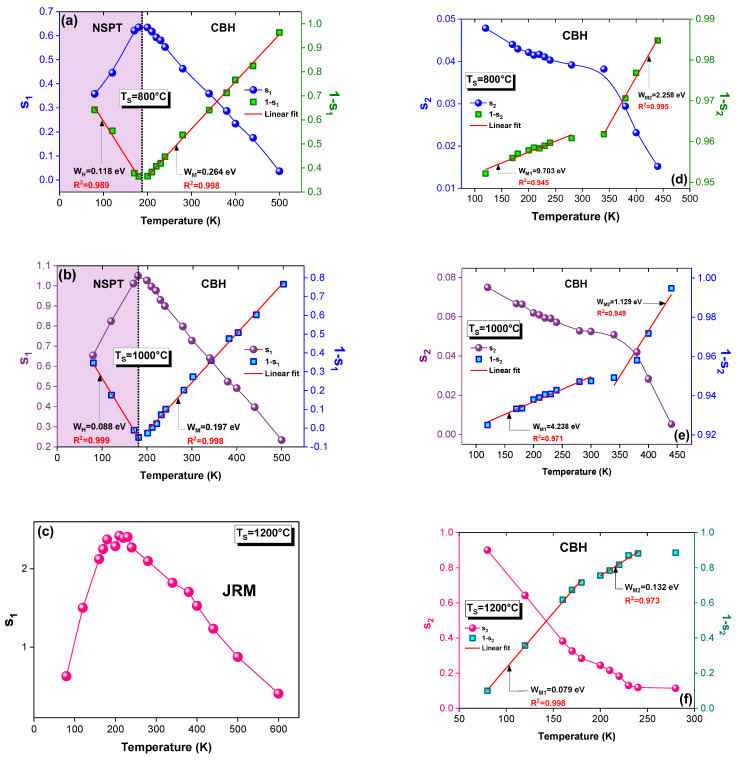
Variation in the frequency exponents (**a**–**c**) ‘s_1_’ and (**d**–**f**) ‘s_2_′ with temperature for the La_0.9_Sr_0.1_MnO_3_ system sintered at (**a**,**d**) 800, (**b**,**e**) 1000 and (**c**,**f**) 1200 °C.

**Figure 3 materials-15-03683-f003:**
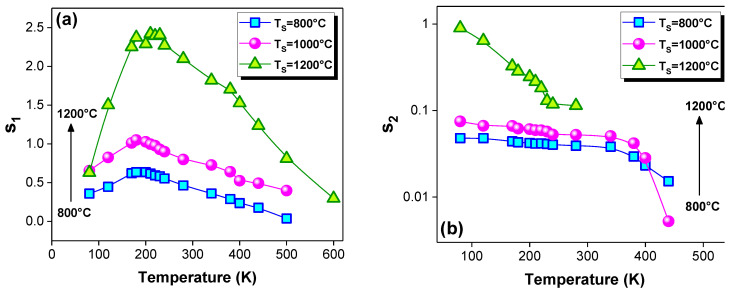
Temperature dependences of (**a**) ‘s_1_’ and (**b**) ‘s_2_’ at different sintering temperatures.

**Figure 4 materials-15-03683-f004:**
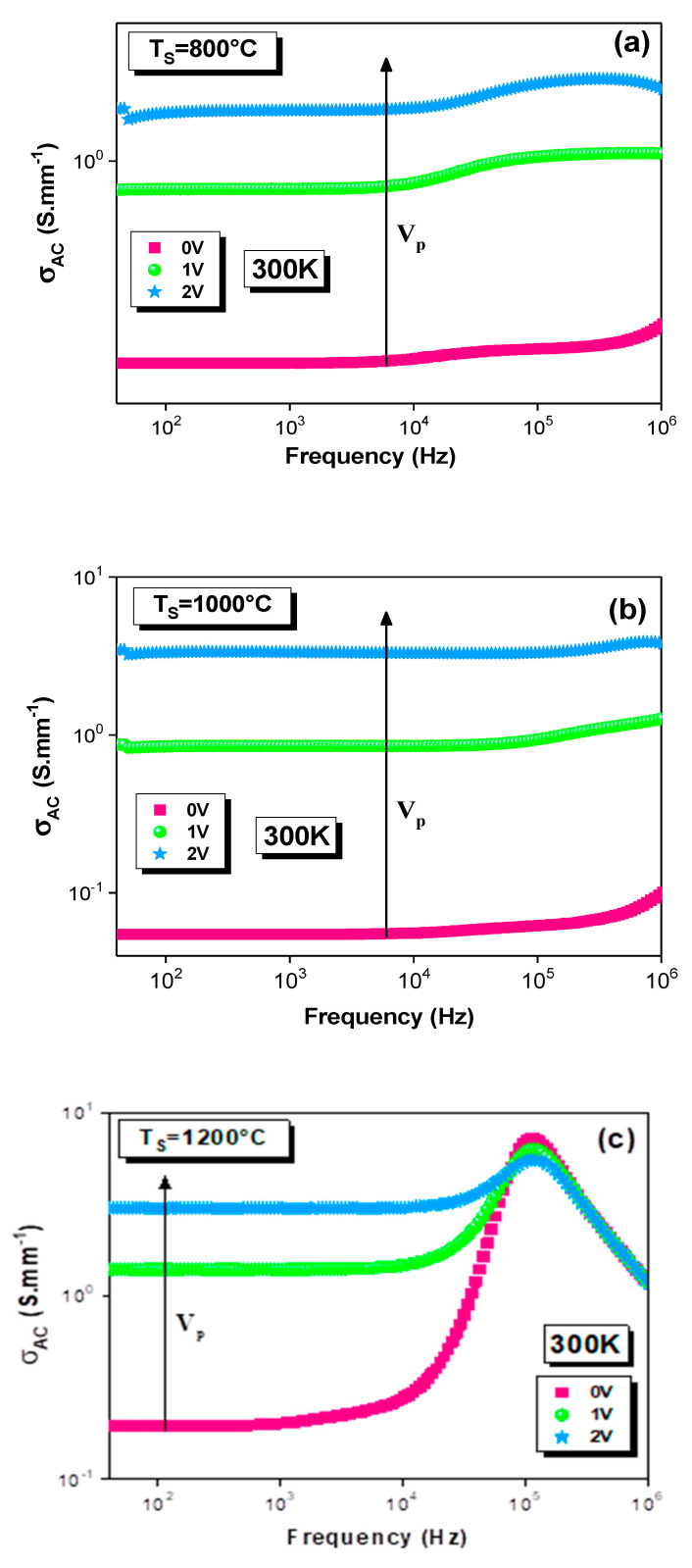
Frequency dependences of the electrical conductivity (σ) of the La_0.9_Sr_0.1_MnO_3_ material sintered at (**a**) 800, (**b**) 1000 and (**c**) 1200 °C at room temperature under electrical polarization with an applied DC biases of Vp = 0, 1 and 2 V.

**Figure 5 materials-15-03683-f005:**
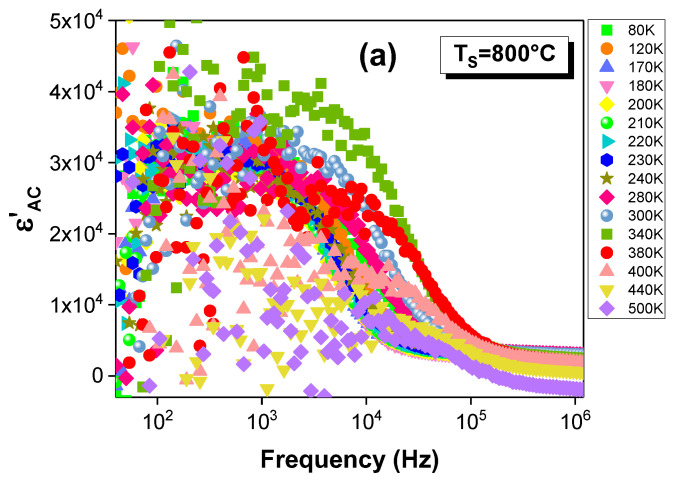
Spectra of the real part of the relative dielectric permittivity (ε′) at different temperatures for the La_0.9_Sr_0.1_MnO_3_ system sintered at (**a**) 800, (**b**) 1000 and (**c**) 1200 °C. (**d**) Schematic representation of the sample structure.

**Figure 6 materials-15-03683-f006:**
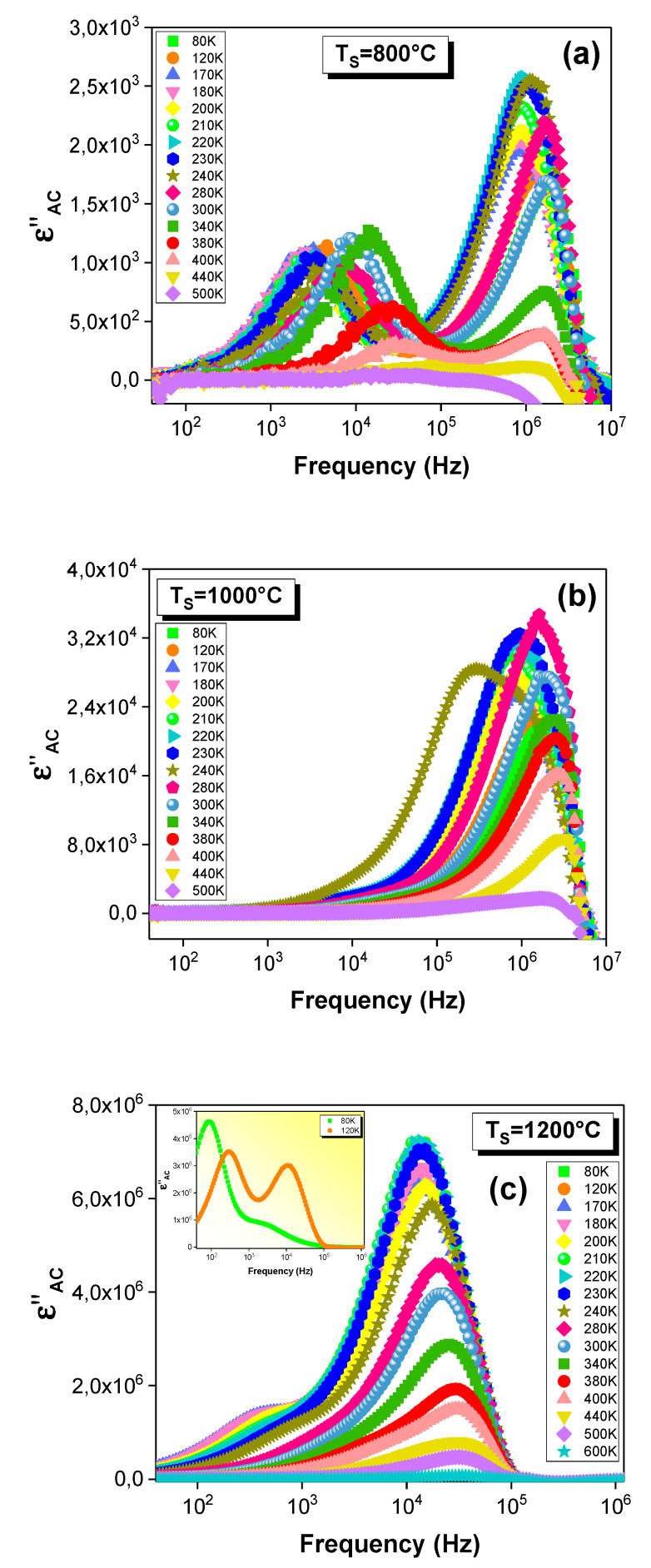
Spectra of the imaginary part of the relative dielectric permittivity (ε″) at different temperatures for the La_0.9_Sr_0.1_MnO_3_ system sintered at (**a**) 800, (**b**) 1000 and (**c**) 1200 °C.

**Figure 7 materials-15-03683-f007:**
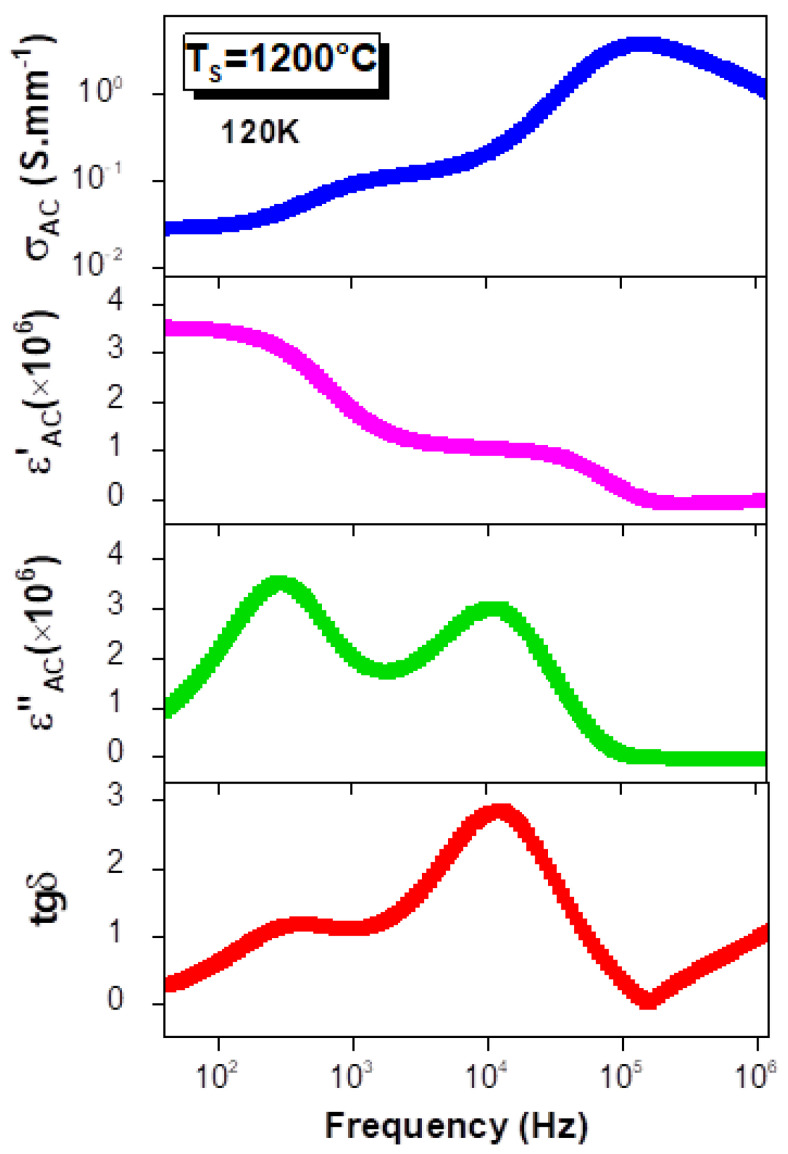
Frequency dependences of AC conductivity, and real and imaginary parts of dielectric permittivity and dielectric loss tangent for the La_0.9_Sr_0.1_MnO_3_ compound sintered at 1200 °C and performed at 120 K.

**Figure 8 materials-15-03683-f008:**
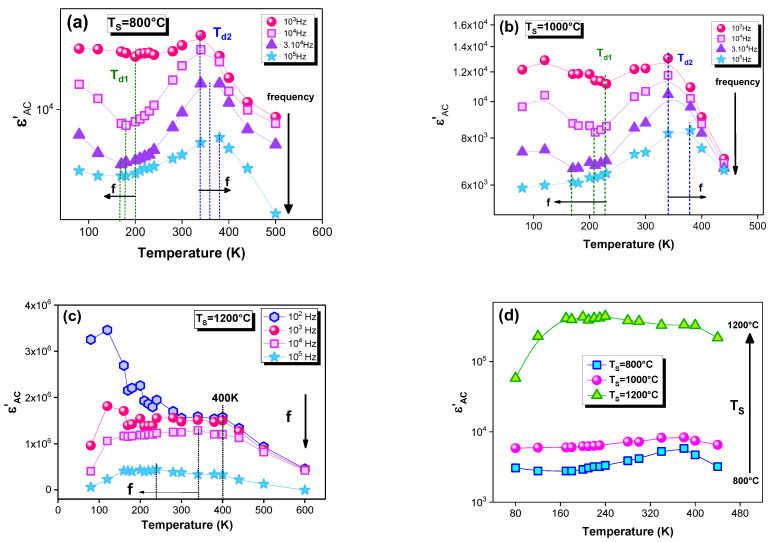
Temperature dependences of the dielectric constant (ε′) at different frequencies for the La_0.9_Sr_0.1_MnO_3_ compound sintered at (**a**) 800, (**b**) 1000 and (**c**) 1200 °C. (**d**) Evolution of the dielectric constant (ε′) with temperature at different sintering temperatures (TS).

**Figure 9 materials-15-03683-f009:**
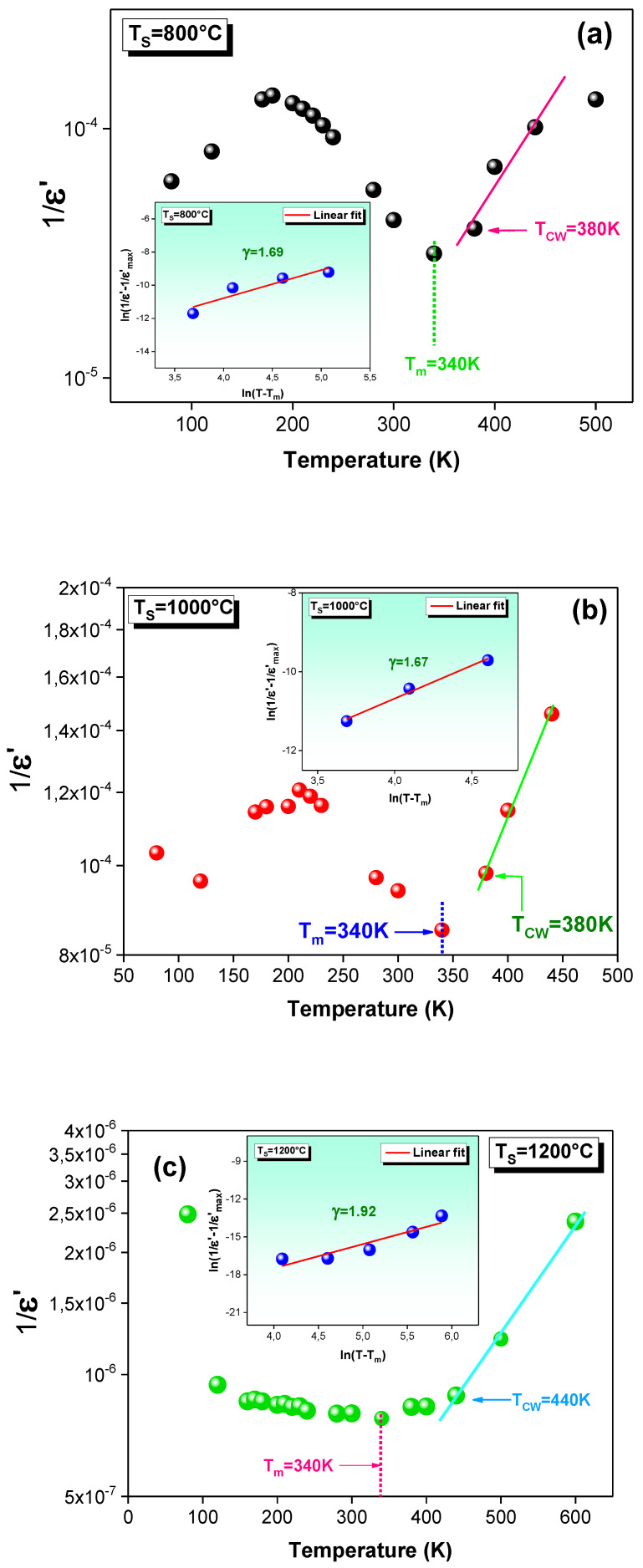
Evolution of the inverse of the dielectric constant (1/ε′) as a function of temperature for the La_0.9_Sr_0.1_MnO_3_ system sintered at (**a**) 800, (**b**) 1000 and (**c**) 1200 °C. The insets are the plots of ln((1/ε′) − (1/ε′max)) against ln(T − Tm) for each sample.

**Figure 10 materials-15-03683-f010:**
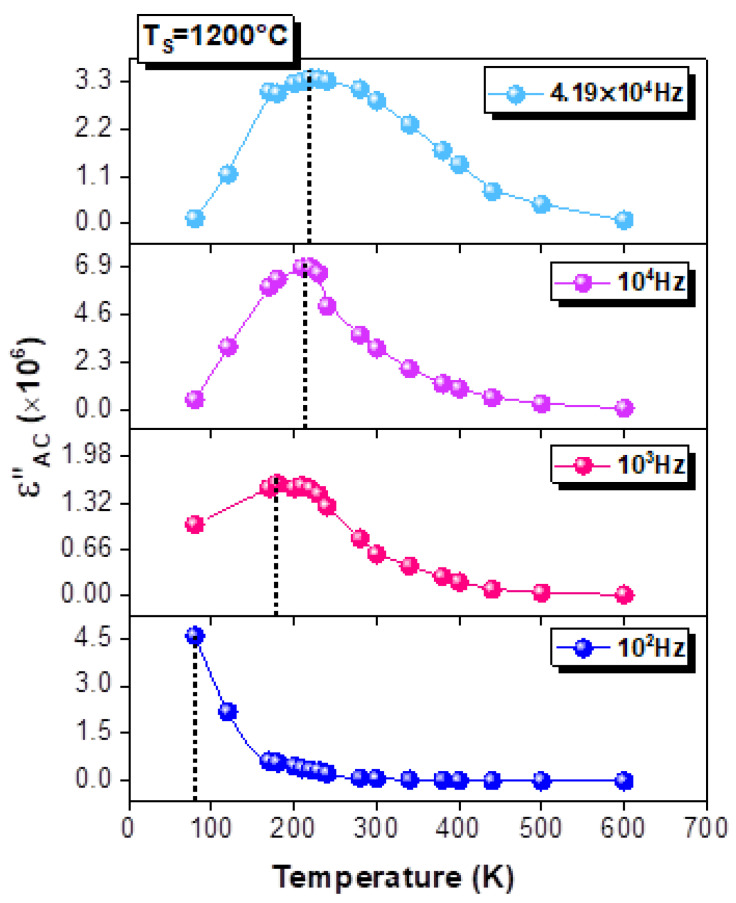
Temperature dependences of the imaginary part of the relative dielectric permittivity (ε″) at different frequencies for the La_0.9_Sr_0.1_MnO_3_ compound sintered at 1200 °C.

## Data Availability

Not applicable.

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
