# Peer review of "Effect of Sintering Temperature and Polarization on the Dielectric and Electrical Properties of La_0.9_Sr_0.1_MnO_3_ Manganite in Alternating Current"

_materials, 2022, doi:10.3390/ma15103683_

Round 1

Reviewer 1 Report

The article by Wired Hizi et al. presents the results of a study of the electrical and dielectric properties of the La0.9Sr0.1MnO3 manganite compound sintered at three different temperatures. It is shown that as the sintering temperature increases, hopping and tunneling are the dominant processes of transfer in the material. In addition, with an increase in the sintering temperature, the dielectric constant increases significantly. The results of the study indicate that such a material is suitable for microelectronic devices. The results are presented in scientific language and confirmed in the study. I think that the materials of the article will be of interest to the readers of Materials. However, there are some points that should be improved in the article: 1. Samples were made from synthesized powders to study electrical properties. How did the porosity and size of these samples affect the properties under study? 2. The quality of drawings should be improved. The designations on them are difficult to read. 3. The permittivity depends on the particle size. The authors should provide the results of particle size measurements.

Author Response

Response:

1/There are a number of factors that must be taken into account regarding electrical and thermal properties. At first glance, electrical connectivity between grains will be influenced by the quality of grain boundaries and the area of contact throughout the samples. This means that percolation must be analyzed, as it is related to porosity within the samples. This means that percolation must be analyzed, as it is related to porosity within the samples. Porosity should decrease thermal conductivity for similar and other reasons, since the presence of air in the pores adds an "insulation" character to the sample that you may be measuring.

2/ According to the reviewer recommendations, the quality of figures is improved

3/ The crystallite size for the sample annealed at 800, 1000 and 1200 estimated from X-ray are found to be 75, 85.01, and 94.9 nm respectively. It is found that crystallite size increases with the increase in annealing temperature, which is in good agreement with obtained results

Reviewer 2 Report

The paper presents a work regarding a material, which could be used for microelectronic devices - La0.9Sr0.1MnO3 manganite. The authors analyzed the effect of sintering temperature and polarization on the dielectric and electrical properties of the compound.

The results obtained are interesting and promising. In my opinion, the manuscript may be acceptable for publication in the Materials after moderate. 

Сomments:

  1. The Materials and Methods section requires additional informations:
    - Dimensions of the tested samples.
    - The number of tested samples / number of tests is not given. What is the repeatability of the test results, what is the error of the measurements, are the obtained results representative?
  2. In the lines 86 and 88, the space group is lacking.
  3. In the Supplementary Materials section, X-ray diffraction patterns of the samples should be presented in order to confim, the samples composition.
  4. Some pictures should be reformatted to improve readability. Some pictures should be increased, sunce the numbers/values are illegible
  5. Figure 5 a/b
    The scattered values at the lower temperatures should be explained. The reproducibility of the data should be confirmed/verified.
  6. The amount of the references is pretty large for research paper. Could it possibly be decreased?
  7. There is a considerable amount of typos such as lacking spaces, lacking of proper indexes in the formulas etc. Please prepare a literature review, body text and the refernces list of a paper according to the guidelines of the Materials journal and MDPI Publisher.

Author Response

Response:

1/ In order to prepare our samples for carrying out electrical measurements, we have to put them in the form of electric dipoles. For this and in the first place, we put our powder into pellets using a hydraulic press with uniaxial pressing.

we tested several pellets of very small size and each time we made the measurements in order to obtain a reproducibility of the measurements the error of the measurements each time very small. and we get good results.

2/  are added in the texte

3/ In this paper we have stady the électrical and dielectrical propriety of the La0.9Sr0.1MnO3 materials  and for deails for XRD please see the refernce 48. Thank you.

4/ According to the reviewer recommendations, the quality of figures is improved.

5/ The variation of the real part of the dielectric permittivity (ε') with frequency at different temperatures for the compound La0.9Sr0.1MnO3 (Tr= 800, 1000 and 1200°C) is shown in Fig. 5 (a-d) . Its behavior with temperature and frequency has been frequently spotted in perovskite materials. Indeed, at low frequencies, ε' exhibits practically constant and very high values for all samples. Then, it decreases with increasing frequency and shows a dispersive behavior at high frequencies where the values of ε' coincide with all explored temperatures. This behavior can be related to the presence of different types of polarization in the studied samples such as electron, ion, dipole and interfacial polarization. At low frequencies, all four types contribute to the total polarization. Indeed, the interfacial and dipole type polarizations strongly depend on the temperature, which explains the temperature dependence of ε'. Therefore, the contribution of interfacial and dipole polarizations dominates in the low frequency domain. These two polarization processes are relaxation phenomena. Dielectric relaxation is the seat of a delay in the response of a system under an external electrical excitation. However, the bias mechanisms tend to follow the direction of the applied electric field in this frequency range. Therefore, the maximum value of ε' is reached. Then, the rapid decrease of ε' can be explained by the increase in the density of free charge carriers which causes a reduction of these types of polarization with increasing frequency. Also, the difficulties encountered by the dipolar material to follow the direction of the electric field can explain this decrease.

6/ we used a lot of references because the results are very interesting and need a lot of explanation.

7/ I have corrected the typos that I encountered in the text. thank you.

Reviewer 3 Report

This is a nice paper presenting dielectric and electrical properties of La0.9Sr0.1MnO3 manganite. I notice that this paper is important especially in the basis on academic viewpoint. At this state the manuscript is not suitable for publication and major revision is needed, as justified in the following points:

  1. In the Experimental details section, it is noticed that the La0.9Sr0.1MnO3 powder is compressed into pellets using a hydraulic press. Please furnish details about this procedure (e.g. the pressure utilized for pellets forming, diameter and thickness of pellets).

  1. Why silver film was employed as electrode? Could silver ions affect the dielectric properties of La0.9Sr0.1MnO3 material? If is possible, please specify the thickness of the silver film deposited onto both sides of each pellet.

  1. Figure 1 emphasizes the frequency dependences of AC-conductivity at various temperature values. However, for a clearer view of AC-conductivity, please add (at least in a supplementary file) the temperature dependences of AC-conductivity at various frequencies.

  1. As follows from Figure 4, the electrical conductivity increases notably under electrical polarization, when the DC-bias is applied between 1 V and 2 V. Also, the amplitude of the alternating electrical field employed for dielectric measurements may affect the electrical conductivity of the material. The authors should mention the AC voltage of the external electrical field selected for dielectric measurements.

  1. In this manuscript, the electrical and dielectric parameters of La0.9Sr0.1MnO3 compound are evaluated as function of frequency and temperature. However, no clear numerical values are specified.

- For example, as presented in Figure 4, the dielectric constant is strongly affected by the interfacial polarization. In that way, it becomes clear that the region of frequency where the dielectric constant decreases slightly with frequency, the ε’ value can be considered as pure dielectric constant of the material which is not affected by other phenomena (in this frequency region, the MW process has no/minor effect). The authors should retrieve the numerical values of ε’ retrieved at high frequencies (100 kHz or 1 MHz). An example of publication about this procedure is Polymer 203 (2020) 122785.

- On the other hand, the DC-conductivity is also an important parameter that should be clearly achieved in respect to the practical applications. Therefore, I consider that the impedance plots -Z″ vs. Z′ need to be included. The impedance spectra may be processed by models (e.g. Cole equation) to determine the numerical values of DC-conductivity. Some of relevant articles may be considered, such as: ACS Applied Polymer Materials 3, 4869-4878 (2021) and Cellulose 28, 843-854 (2021). Hence, a complete examination based on dielectric spectroscopy measurements will be achieved. Please comment on this important point and include your answers and experimental results in the revision.

Author Response

Response

1/ an2/  In order to prepare our samples for carrying out electrical measurements, we have to put them in the form of electric dipoles. For this and in the first place, we put our powder into pellets using a hydraulic press with uniaxial pressing. Figure  shows an actual image of the hydraulic press. This operation consists of compacting the particles of the powder and increasing the contact points between the grains.

In a second place, we deposited a thin metallic layer of silver (our choice) on each of two faces of the pellets using a vacuum thermal evaporator. This vacuum is completed by a device which comprises two pumps. A pump provides a primary vacuum and the second causes a secondary vacuum which is suspended in the center of the tubular enclosure. This technique consists of the sublimation of silver (material to be deposited) by the Joule effect. Indeed, the samples are placed in the vacuum evaporator on a circular mask. We also placed a tungsten crucible in an enclosure where there is a secondary vacuum of about 10-5 mbar in order to avoid all kinds of contamination and oxidation, on which is placed a silver ball. It is necessary that the evaporation temperature of silver should be lower than the melting temperature of the crucible. Tungsten has a fairly high melting temperature (3422°C). This, makes it the right choice. Thus, the silver begins to evaporate in the chamber and to condense on the exposed parts of the samples when a large current is circulated in the crucible. Indeed, the vacuum favors the vertical propagation of the ejected flow without deviation. The quartz balance controls the thickness of the silver layer deposited (200 nanometers). The samples then take the configuration of a planar capacitor.

3/ the silver used as electrode it does not affect the properties of our materials.

The quartz balance controls the thickness of the silver layer deposited (200 nanometers).

the silver used as electrode it does not affect the properties of our materials.

4/ Al value are added in the figures.

5/ I have rectify some mistak

I added this ref in 66.Dominik Schwaiger, Wiebke Lohstroh, and Peter Müller-Buschbaum, Investigation of Molecular Dynamics of a PTB7:PCBM Polymer Blend with Quasi-Elastic Neutron Scattering, ACS Appl. Polym. Mater.(2020), 2, 9, 3797–3804.

And the second paper in ref 69. Jankowska, I.A., Pogorzelec-Glaser, K., Ławniczak, P. et al. New liquid-free proton conductive nanocomposite based on imidazole-functionalized cellulose nanofibers. Cellulose 28, 843–854 (2021).

Round 2

Reviewer 2 Report

Author has properly addressed the concerns from the referee. 

Author Response

Dear sir

We have modify the conlusion

Thank you

Reviewer 3 Report

The manuscript was poorly revised and the authors did not furnish the corrections in accordance with my suggestions. I recommend the authors to response to all comments and to include them into a detailed list of answers. Also, please include the answers and experimental results in the revision version of the manuscript.

Author Response

Please sir find enclosed the response for all of the questions.

Reviwer3

  1. In the Experimental details section, it is noticed that the La0.9Sr0.1MnO3 powder is compressed into pellets using a hydraulic press. Please furnish details about this procedure (e.g. the pressure utilized for pellets forming, diameter and thickness of pellets).

In Experimental details we have added

-La0.9Sr0.1MnO3 material was synthesized using the citrate-gel method. In fact stoichiometric amounts of the nitrate precursor reagents La(NO3)3 6H2O, Mn (NO3)2 4H2O and Sr(NO3)2 were dissolved in water and mixed with ethylene glycol and citric acid, forming a stable solution. The molar ratio metal:citric acid was 1:1. The solution was then heated on a thermal plate under constant sintering at 80 °C to eliminate the excess water and obtain a viscous gel. The gel was dried at 130 °C and then calcinated at 600 °C for 12 h.

-The powders obtained were ground then pressed into pellets a few millimeters thick (~ 2 mm) and with a diameter of around 1 cm under 10 tonnes/cm2. These pellets were sintered at 1000°C for 24h. In order to obtain the pure crystalline phases, the pellets obtained were finely ground and the resulting powders underwent some further cycles of grinding, pelleting and sintering at 1100°C.

The sample in the form of a tablet is fixed on the cryostat by the silver lacquer. Then, the wires fixed on each surface of the patch are connected to the electrodes of the cryostat. Then the connection of the wires of the cryostat to those of the impedance bridge was ensured by coaxial cables to conduct the signal and protect it from noise. The cryostat is connected to the temperature controller and the vacuum pump. The measurement is started according to the frequency. The following parameters are adjusted: the couples to be measured Cp-G, Cp-D and Z'-Z", the frequency from 40 Hz to 110 MHz, the excitation voltage at 50 mV, with or without bias voltage. to make sure you have a good contact, it is advisable to carry out a test at room temperature. Then start pumping (~ 30 min) until the vacuum is established. Set the starting temperature in the regulator. Liquid nitrogen is gradually poured into the cryostat (inner tank of the cryostat). When the temperature reaches 80 K, measurements are started and recorded each time the temperature is stabilized. The temperature is increased from 80 K to 700 K with a step of 20 K.

  1. Why silver film was employed as electrode? Could silver ions affect the dielectric properties of La0.9Sr0.1MnO3 material? If is possible, please specify the thickness of the silver film deposited onto both sides of each pellet.

The silver lacquer is painted directly on the faces of the specimens in order to form a thin metallic layer in close contact with the rock. The electrical contact is thus ensured as well as possible. However, these electrodes are not completely non-polarizable, and show a tendency to oxidation on contact with the sample saturated with saline solutions: they therefore require care to be taken to ensure that they are in good condition. A preliminary study on sandstones from Fontainebleau showed that silver lacquer makes it possible to create reliable electrodes. They will be used for samples not containing smectite.. In the case of samples containing smectite-type clays, which react in contact with lacquer, you can use stainless steel electrodes.

The silver  serves as an electrical connector. A pellet of about 0.503 cm2 surface and about 0.14 cm thickness was used for the electrical measurements. The pellet disc was coated with Ag paste to ensure good electrical contact.

  1. Figure 1 emphasizes the frequency dependences of AC-conductivity at various temperature values. However, for a clearer view of AC-conductivity, please add (at least in a supplementary file) the temperature dependences of AC-conductivity at various frequencies.

The experimental device used for the dielectric characterization of our samples. It includes an Agilent 4294A impedance analyzer which measures impedance (or associated quantities) in the 40 Hz – 110 MHz frequency range under different excitation and bias voltage conditions. For temperature studies, the sample is placed in a liquid nitrogen cryostat which allows the temperature to vary within a range extending from 80 K to 700 K. The measurements are transmitted to a microcomputer for processing.

thank you for your understanding and help in improving the paper.

The temperature dependence of AC-conductivity, at some well chosen frequencies, is shown in this Figure. It is observed that the AC-conductivity increases when increasing temperature for each curve. Such behavior proves the thermal activation of AC-conductivity which follows the relation :

where σ0 is a pre-exponential factor, kB is the Boltzmann constant and EAC is the conductivity activation energy sAC

  1. As follows from Figure 4, the electrical conductivity increases notably under electrical polarization, when the DC-bias is applied between 1 V and 2 V. Also, the amplitude of the alternating electrical field employed for dielectric measurements may affect the electrical conductivity of the material. The authors should mention the AC voltage of the external electrical field selected for dielectric measurements.

-The net effect of temperature is observed over the entire frequency range for each spectrum of sAC. This result proves that the conduction process in the studied samples is thermally activated. These spectra are composed of two regions of different frequencies depending on their behavior. A plateau appears for each spectrum at low frequencies, in which the conductivity is nearly frequency independent and thermally activated. This region is associated with the long-range translational motion of ions. During the annealing process, the frequency independence region extends over less frequency range. Indeed, when the annealing temperature increases, the growth of the curves of sAC occurs in lower frequencies with more intense slopes. Thus the jump frequency decreases with Tr. This is well visualized for Tr=800, 1000 and 1200 °C. While for each sample, the jump frequency increases under the effect of temperature. Thus, charge carriers take less time to move. As a result, their mobility increases which is manifested in the increase in conductivity under the effect of temperature.

-In the second frequency region, the electrical conductivity increases with the frequency showing on the one hand the potential intervention of different conduction mechanisms. This behavior is due to the creation of new conduction sites and the release of trapped charge carriers generated by the disturbance of the crystal under the effect of the frequency. On the other hand, this evolution confirms that the jump process is the most likely to describe electrical conduction. The value of the frequency at which the slope changes, in each spectrum and for each temperature curve, is the jump frequency νH. In fact, the increase in sAC is established when the jump of the charge carriers does not follow the applied field beyond a certain high frequency (νH).

- The experimental bench for electrical characterization by impedance spectroscopy is available in our Laboratory and is composed of:

❶ An Agilent 4294 A impedance bridge.

❷ A liquid nitrogen cryostat, VPF-800, allowing temperature variation from 80 to 700 K.

❸ A computer, ensuring data acquisition and storage.

❹ A Lakeshore 335 temperature controller, allowing temperature measurement and regulation at the sample level.

❺ A vacuum pump.

Impedance spectroscopy is a powerful technique in the field of electrical characterization. It is more used for solid materials such as mixed oxides of the perovskite type, dielectrics and ceramics. The principle consists in disturbing the material by a sinusoidal signal (tension) of weak amplitude and variable frequency will be crossed by an electric current to determine the intrinsic properties of materials. The ratio obtained between the voltage and the current is the impedance. The measurement of impedance as a function of frequency is impedance spectroscopy.

just to explain to you sir that we use a variable voltage and not an external field.

thank you for your understanding sir.

5/ a/Juste  here sir our objective and to follow the evolution of dielectric parameters of La0.9Sr0.1MnO3  according to the frequency because it is very important to see its behavior with the frequency. It can give us an idea about the different types of polarization in the studied samples such as electronic, ionic, dipole and interfacial polarization.

b/ On the other hand, the DC-conductivity is also an important parameter that should be clearly achieved in respect to the practical applications. Therefore, I consider that the impedance plots -Z″ vs. Z′ need to be included. The impedance spectra may be processed by models (e.g. Cole equation) to determine the numerical values of DC-conductivity. Some of relevant articles may be considered, such as: ACS Applied Polymer Materials 3, 4869-4878 (2021) and Cellulose 28, 843-854 (2021). Hence, a complete examination based on dielectric spectroscopy measurements will be achieved. Please comment on this important point and include your answers and experimental results in the revision.

This the point of the impedance plots -Z″ vs. Z′ was discussed in an article recently published in catalyst by the same group

« Impact of Sintering Temperature on the Electrical Properties of La0.9Sr0.1MnO3 Manganite » by

W Hizi. Et al, Catalysts 12 (3), 340(2022)

c/I added this ref in 66.Dominik Schwaiger, Wiebke Lohstroh, and Peter Müller-Buschbaum, Investigation of Molecular Dynamics of a PTB7:PCBM Polymer Blend with Quasi-Elastic Neutron Scattering, ACS Appl. Polym. Mater.(2020), 2, 9, 3797–3804.

And the second paper in ref 69. Jankowska, I.A., Pogorzelec-Glaser, K., Ławniczak, P. et al. New liquid-free proton conductive nanocomposite based on imidazole-functionalized cellulose nanofibers. Cellulose 28, 843–854 (2021).

We hope that these revisions give greater clarity to the points being made. We appreciate again the comments from the reviewers and thank you for reviewing our manuscript.
